# Hemostatic Enhancement via Chitosan Is Independent of Classical Clotting Pathways—A Quantitative Study

**DOI:** 10.3390/polym12102391

**Published:** 2020-10-17

**Authors:** Kuan-Yu Chen, Yen-Cheng Chen, Tzu-Hsin Lin, Cheng-Ying Yang, Ya-Wen Kuo, U. Lei

**Affiliations:** 1Institute of Applied Mechanics, National Taiwan University, Taipei 10617, Taiwan; d98543006@ntu.edu.tw (K.-Y.C.); r03543026@ntu.edu.tw (Y.-C.C.); 2Department of Traumatology, National Taiwan University Hospital, Taipei 10002, Taiwan; Jsl555@gmail.com; 3Department of Multimedia and Game Science, Chung Chou University of Science and Technology, Changhua 51003, Taiwan; cyyang1234@gmail.com; 4CoreLeader Biotech Co., Ltd., 19F., No. 100, Sec. 1 Xintai 5th Rd., Xizhi Dist., New Taipei City 22102, Taiwan; ywk701@gmail.com

**Keywords:** chitosan, classical clotting pathways, flow-through device, hemostasis, wound gauzes

## Abstract

Hemostasis is a process causing bleeding to stop, and it is known from the literature that hemostasis can be enhanced using chitosan on wound gauzes. We proposed here a continuous flow-through device, with the test blood flowing through the gauze sample at a constant flow rate and the pressure drop across the gauze measured, for assessing the hemostatic performance of the gauze. Experiments were performed using the device with both whole blood and washed blood (with clotting factors and platelets removed from the whole blood), and their results agree with each other within 10% discrepancy, indicating quantitatively that hemostatic enhancement via chitosan is essentially independent of classical clotting pathways, which was demonstrated qualitatively through animal tests in the literature. The proposed device and method can be applied for evaluating quantitatively the hemostatic performance of various gauzes in a flowing blood environment (in comparison with static tests) with less test blood (20–60% less, in comparison with that of a flow-through device driven by a constant pressure gradient), and are thus, helpful for designing better wound gauzes. In particular, it is effective to enhance the hemostatic performance further (additional 30%) through acidification (changing the amino group to the ammonium group) of the gauze for chitosan-based wound gauzes.

## 1. Introduction

Chitosan is a linear copolymer consisting of units of acetyl glucosamine (C_8_H_15_NO_6_) and glucosamine (C_6_H_13_NO_5_) [1]. It is a material with a variety of biomedical applications, as it is non-toxic, antibacterial, biocompatible, and biodegradable, and possesses high water absorption capacity and reactive functional groups [2,3,4]. Here, we focus on the hemostatic characteristics of chitosan-based wound gauzes [5,6].

Hemostasis is a process causing bleeding to stop. The conventional practice for stopping the bleeding is to apply an absorbent gauze dressing (cotton gauzes, for example) to the wound, in company with an applied pressure opposite to the blood flow through packing or using tourniquets. However, there are many situations where bleeding is still uncontrolled, such as that due to military trauma; several advanced hemostatic products [6,7,8] were thus developed for use in cases of severe uncontrolled bleeding. Among those products, chitosan-based gauze/dressing is of particular interest because it can arrest bleeding with clotting dysfunction (without the functions of platelets and coagulation or clotting factors), which was demonstrated qualitatively through animal tests [9,10] and medical treatments in practice [11,12,13]. Furthermore, chitosan-based gauze/dressing possesses also many beneficial biomedical characteristics associated with wound treatment (antifungal, bactericidal, permeable to oxygen, for example) [4,5,14], and two chitosan-based gauzes were recommended as the hemostatic dressing of choice by the Tactical Combat Casualty Care (TCCC) guidelines [6]. In addition to military trauma, chitosan-based hemostatic products were also applied to uncontrolled external hemorrhage in civilian emergency medical services (EMS) systems [11], to surgery operations [12,13], and to backcountry care [15]. Hemostatic efficacy can be enhanced by further crosslinking fibrin to the chitosan gauzes [16].

Although there are literature showing that chitosan can induce platelet adhesion and aggregation, and activate several coagulation factors or enhance their expressions [17,18], there is even more evidence from animal tests [9,10], surgery operations [12,13], and static experimental tests [19,20], subject to clotting dysfunction, showing that hemostatic enhancement via chitosan is independent of classical clotting pathways. Because of the opposite charges between chitosan and red blood cells (RBCs), it is generally claimed in the literature [6,15], based mainly on observation, that chitosan attracts and crosslinks with red blood cells (RBCs) to form a “mucoadhesive barrier” at the wound for stopping the bleeding. The accumulation of RBCs on the chitosan gauze is a prerequisite for the formation of the “mucoadhesive barrier”, and how the RBCs accumulate is of particular interest. Through some qualitative RBCs aggregation experiments and some supporting quantitative measurements of zeta potentials of RBCs and pHs of blood–chitosan mixtures, Chen et al. [21] proposed that there are two types of RBC accumulation associated with chitosan objects: the direct adhesion of RBCs on chitosan objects and the aggregation of RBCs among themselves next to the objects. Direct adhesion of RBCs on the chitosan object could be understood because of the Coulomb force associated with negative charges on RBCs (the COO^−^ molecules) [22] and positive charges (the NH_3_^+^ molecules) on chitosan [2]. However, the number of RBCs adhered directly to chitosan is limited as the charges on chitosan are finite and the Coulomb force decreases inversely with the square of the distance; it is thus expected that one or, at most, several layers of RBCs, can adhere directly to the chitosan surface. On the other hand, the mesh sizes of the gauzes, of order of 1 mm, are of two orders greater than the size of RBCs and tens or hundreds of RBC layers are required for blocking the grid space of the gauzes. Thus, the ability of the aggregation of RBCs themselves in the space of the gauze mesh is crucially associated with hemostatic enhancement, as it is a plausible way for forming the “mucoadhesive barrier”. Aggregation of RBCs (forming layer-by-layer structures) occurs experimentally next to chitosan yarns, strips, and gauzes in both static and flowing environments [21]. The “mucoadhesive barrier” is actually a layer-by-layer structure of RBCs reinforced via some biological, chemical, and physical means, and grows to a size comparative to the mesh size of the gauze. Chen et al. [21] proposed that the formation of the RBC layer structure next to chitosan objects is due to the reduction in the repulsive electric double layer force between RBCs, with the attractive forces [23,24,25] between RBCs remaining essentially unchanged, under the DLVO (Derjaguin–Landau–Verwey–Overbeek) theory [26]. The reduction in the repulsive electric double layer force among RBCs occurs because of the association of H^+^ (deprotonated from chitosan) with COO^−^ on RBC membrane.

The discussion in the above paragraph suggests that the primary mechanism of hemostatic enhancement associated with chitosan could probably be of mechanical nature, but it is still unknown how large is such a mechanical effect in comparison with the effects associated with classical clotting pathways for hemostasis. It would be helpful for a thorough understanding of the hemostatic mechanism via chitosan if one could elucidate quantitatively via measurement the hemostatic issue in a controlled flowing blood environment, which is the motivation and also the primary goal of this study. This study is important not just from the academic view point (that the hemostatic enhancement via chitosan is independent of classical clotting pathways), but is also beneficial for designing a better chitosan-based wound gauze, as one can assess the relative hemostatic performance of two different gauzes only when one can quantify them. We will perform experiments using both the whole blood and the washed blood in a proposed flow-through device (called the CFR device) with chitosan-based gauzes, and compare their results. The washed blood defined here is “the washed red blood cells suspended in saline” at a hematocrit similar to that of the whole blood, with platelets and clotting factors removed. On the other hand, the whole blood contains platelets and all clotting factors intrinsic to human blood. We will also perform experiments using other gauzes (cotton, rayon, and non-woven gauzes) for comparison.

As for the testing methods for hemostatic performance, although the animal test is the gold standard in the testing of gauzes/dressings and is required for the final assessment of an important new dressing, it is not suitable for studying the detailed hemostatic characteristics of materials and assessing efficacy among different gauzes/dressings. Most of the current studies on the hemostatic characteristics of chitosan [14,16,17,18,19,20,27] employ standard coagulation tests, blood clotting, and platelet adhesion assays (involving the turbid-metric method and optical detection), cytometry, and scanning electron microscopy. The measurements using the above methods do provide us with important information for understanding the hemostatic characteristics of chitosan, but are not convenient for discriminating the efficacy among different gauzes and most of them perform tests in a static environment. On the other hand, Jesty et al. [28] proposed a flow-through method for assessing in vitro the active hemostatic properties of gauzes, which is convenient for comparing the performance of various gauzes. Test fluid flows through a chamber containing gauze/dressing elements in the flow-through device of reference [28] under an applied pressure gradient, with the amount of the fluid masses through the gauze elements measured, and is employed for characterizing the hemostatic performance. If the flow-through fluid mass reduces for gauze sample A in comparison with that for gauze sample B in a given time interval, the performance of sample A is better than that of B. The device of reference [28] is denoted here as the CPG device, as it is driven by an applied Constant Pressure Gradient. It enables us to study the hemostatic phenomenon in a flowing environment, which provides us a more realistic situation than that in a coagulation test or a clotting assay, as stated above. 

We applied the CPG device of reference [28] in the present study, and indeed, we have performed hemostatic experiments using a CPG device, as shown in Appendix A. However, the size or the required space of the CPG device is not small, and a considerable amount of test fluid is required. It would be helpful if an alternative device, say, a smaller flow-through device with less test fluid, is available, which is the second goal of the present study. The proposed alternative device allows the test fluid to flow through the gauze elements at a Constant Flow Rate (called the CFR device here), with the pressure drop across the gauze elements measured for characterizing the hemostatic properties. Jesty et al. [28] mentioned the idea of the CFR device, but the device was not implemented. 

## 2. Materials and Methods

### 2.1. Flow Device

The CFR device is the device employed in reference [21] with modification, as shown in Figure 1a. We installed two glass tubes (served as a monometer) additionally here for measuring the pressure drop across the gauze layers. The CFR device was molded with polydimethylsiloxane (PDMS) on a glass (or PDMS) substrate. It is essentially a rectangular straight channel passing through a gauze chamber, which is a rectangular box as shown in the figure. Four layers of gauzes (each with size of 9 mm × 5 mm) were stacked and inserted into the chamber for testing. More layers could be inserted for providing a larger pressure drop. A typical chitosan-based gauze is shown in Figure 1b as an illustration, with its typical mesh size shown in Figure 1c. Test fluid was forced to flow through the gauze chamber using a syringe pump at a given constant flow rate, which was set as 25.2 mL/h for the measurements presented here. Other volume flow rates were also tested, with essentially the same qualitative results among different gauzes. The pressure drop measured is the indicator for the hemostatic efficiency. A higher pressure drop indicates a better hemostatic performance, because the drag on the fluid (and thus, the pressure drop) increases as a result of the degree of interaction between blood and gauze material.

### 2.2. Gauzes

Different gauzes were tested in this study. They are chitosan gauzes with acidification (-A1 and -A2) and without acidification (-NA1), cotton gauze, non-woven gauze, and rayon gauze, as listed in Table 1. The length scales of the mesh are in general of two orders greater than the size of red blood cells. The cotton and non-woven gauzes were purchased commercially (CSD Company, Changhua City, Taiwan), and the rayon and chitosan gauzes were obtained from CoreLeader Biotech Co., Ltd., New Taipei City, Taiwan. Rayon is an engineered regenerated cellulose fiber; several rayon fibers are braided into a yarn, and yarns are then woven into rayon gauze. The chitosan gauze—A1 is actually a composite gauze woven with yarns consisting of chitosan and rayon fibers, which is the basic element of a commercial product, HEMO-bandage (CoreLeader Biotech Co., Ltd.), with a base weight of 85 g/m^2^. Detailed information, including the materials and fabrication processes of the composite yarns, are available in reference [29]. The chitosan gauzes (-A1 and -A2) were acidified in an acid alcohol using acetic acid (J. T. Baker, Phillipsburg, NJ, USA) and ethanol solution (Seng Fa Chemical Biotech Co., Ltd., Taiwan), and washed and dried before they were applied for the experiment. Details of the fabrication of the gauzes and bandage, as well as the acidification process of them, were discussed in reference [30]. Here, the “acidification process” refers to changing the amino group (–NH_2_) to the ammonium group (–NH_3_) on chitosan, which is not the deacetylation process of deriving chitosan from chitin. The latter is to replace the acetyl functional group (–CH_3_CO) on chitin by H, forming the amino group on chitosan. In short, the acidification process here is to impart more H^+^ ions on the chitosan gauze. The chitosan gauze—A2 is similar to that of chitosan gauze—A1, but with a larger base weight (166 g/m^2^) and a smaller mesh size. The chitosan gauze—NA1 is similar to that of the chitosan gauze—A1, but without the process of acidification. In addition, the base weight of the chitosan gauze—NA1 is greater than that of the chitosan gauze—A1. The above gauze information is summarized in Table 1.

### 2.3. Blood

Informed consent procedures for blood donations were approved by the National Taiwan University Institutional Review Board on Human Research. Blood was collected from adult volunteers into a vacutainer (Becton Dickinson, yellow cap), which contained sodium polyanethol sulfonate and acid citrate dextrose additives (ACD; 22.0 g/L trisodium citrate, 8.0 g/L citric acid, 24.5 g/L dextrose) as the anticoagulant for preventing blood clotting. The mixture of blood and additives (called whole blood–1, here) behaves like acid, and its pH was measured as 6.6 ± 0.21, which is less than the pH of blood in the human body (around 7.5). The pH of the whole blood can be adjusted to around 7.5 by adding a small amount of NaOH into it. Both the whole blood with average pH 6.6 and 7.52 (called whole blood–2) were employed in the experiment.

Besides the whole blood, washed blood was also employed in this study. The procedures for the preparation of washed blood were as follows [31]. (1) The whole blood–1 was centrifuged at 3000 rpm for 5 min at room temperature. (2) After the spin, three distinct layers of the blood were formed in the tube: the top straw-colored layer of platelet-rich plasma, the middle thin layer with mainly the white blood cells, and the bottom layer with red blood cells (RBCs). Both the top and the middle layers were removed using a transfer pipette. (3) The remaining red blood cells were mixed with normal saline at a 1:1 ratio by volume. The mixture was then centrifuged at 3000 rpm for 5 min at room temperature. After the spin, the supernatant was removed using a transfer pipette. (4) Procedure (3) was repeated three times and the liquid medium above the RBC layer was removed. The remaining RBCs are called washed RBCs. (5) The washed RBCs were mixed with normal saline at volume ratios 4:6 (washed RBCs: saline = 4:6) and 5:5, so that the hematocrits of the resulting mixture were slightly (of several %) less than 40% and 50%, respectively, because there was still some saline adhered to the RBCs. The mixture was called washed blood in this study. The average values of pH were measured as 7.08 and 7.55 for the 4:6 and 5:5 washed bloods, and are called washed blood–1 and washed blood–2, respectively. Using washed RBCs has long been a fundamental technique in scientific research and applications in medicine, adopted in nearly all blood banks and most hospital laboratories, aiming to preserve the RBCs without any other seral substances such as antibodies or globulins to interfere with its coagulation properties. All the above blood samples are summarized in Table 2.

## 3. Results

### 3.1. The Hemostatic Performance of Various Gauzes

Before the constant flow rate (CFR) device is applied for assessing the hemostatic performance of the gauzes, it is required to know the pressure drops across various gauzes (refer to Table 1) resulting solely from the blockage effect of the gauzes associated with fluid mechanics, as a baseline for studying hemostasis associated with other interactions between blood and gauze materials. Here in this paper, the pressure is expressed in terms of the height of the liquid column (H_c_) in the glass tube in Figure 1a, and the pressure drop across the gauzes is the height of the liquid column upstream minus that of the liquid column downstream of the gauze chamber. The actual value of the pressure is *ρ*gH_c_, with *ρ* the density of the test liquid, and g = 9.8 m/s^2^, the value of gravity. A mixture of glycerol (40%) and de-ionized water (60%) [32] was employed as the test fluid for experiment, with the results shown in Figure 2. The viscosity of the above glycerol–water mixture was measured as 4.08 cP, which is close to the viscosity of human whole blood (about 4 cP) with hematocrit around 40% [33]. The pressure drop across a given gauze was measured at different selected times (with a time interval 30 s here) for a given test. The pressure drop across the cotton gauze is 2 mm for all the times in Figure 2. There are few points (7 out of 31) for the chitosan gauze (A1) that the pressure drop is of 0.5 mm higher, but also few points (6 out of 31) that the pressure drop is of 0.5 mm lower for the non-woven and rayon gauzes. Note that the minimum reading (based on the scales of a ruler next to the glass tube) of the height of the liquid column is 0.5 mm in the present CFR device. Thus, the pressure drops across various gauzes in Figure 2 associated with the blockage effect of fluid mechanics are essentially 2 mm of liquid height, irrespectively of the different gauzes in this study. Moreover, the pressure drops for all the four gauzes do not change with time essentially, for a homogeneous liquid such as the glycerol–water mixture here. 

However, if the test fluid is whole blood–1 (see Table 1) instead of the glycerol–water mixture, the pressure drops increase with time for all the four gauzes, as shown in Figure 3. The increases are large in comparison with the pressure drops associated with the blockage effect of fluid mechanics in Figure 2. The average value of pressure drop increased from 2.25 to 10.75 mm for the chitosan gauze—A1, from 2 to 6.38 mm for the cotton gauze, from 2 to 5.13 mm for the non-woven gauze, and from 2 to 3.38 mm for the rayon gauze, as the time changes from 1 to 16 minutes. The averages and error bars in Figure 3 and other figures of this paper are based on five measurements with different blood and gauze samples.

The hemostatic performance for a given gauze is proportional to the pressure drop across it, and the order for a better performance according to Figure 3 is “chitosan gauze—A1 > cotton gauze > non-woven gauze > rayon gauze”, which agrees with the result using a CPG device in Appendix A. The value of the pressure drop at the end of the present test (i.e., at 16 minutes) across the chitosan gauze—A1 is 3.18 times that across the rayon gauze, and 1.68 times that across the conventional medical cotton gauze.

Figure 4 shows the results for different chitosan-based gauzes, with different base weights and with or without acidification. The pressure drop for chitosan gauze—A2 is higher than that for chitosan gauze—A1 because of the higher base weight, as there is a higher amount of chitosan in the A2 gauze. However, the hemostatic performance of the acidified A1 gauze is substantially better than that of the non-acidified NA1 gauze, even though the latter has a higher base weight.

### 3.2. Tests of the Hemostatic Performance of Chitosan Gauzes Using Whole and Washed Bloods

All the results in Figure 3 and Figure 4 are based on whole blood–1. Here, in this subsection, we will study the hemostatic performance of chitosan gauze—A1 using all the four blood samples, including both the whole bloods (including all the clotting factors and platelets) and washed bloods (with clotting factors and platelets removed), in Table 2.

Consider first the results for whole blood–1 and those two washed bloods in Figure 5. They agree with one another within experimental errors, indicating that the hemostatic enhancement associated with chitosan is essentially independent of classical clotting pathways (i.e., without the functions of clotting factors and platelets). The result of whole blood–1 agrees more closely to that of washed blood–2, indicating that the hematocrit of whole blood–1 is more likely to be between 40% and 50%. The average pressure drop for the case with higher hematocrit (washed blood–2) is slightly greater than that with lower hematocrit (washed blood–1), because there are more RBCs, and thus, more interaction between RBCs and chitosan in the case with higher hematocrit. As the pH could affect the interaction between RBCs and chitosan [21], and the pHs for whole blood–1 (6.6) and those two washed bloods (7.07 and 7.55) are different as indicated in Table 2, a further test using whole blood–2 (with pH 7.52) was thus performed and its results are also plotted in Figure 5. The pressure drop for the case at a higher pH (7.52) for whole blood–2 is smaller than that at a lower pH (6.6) for whole blood–1 at an earlier time, but it matches the result of whole blood–1 and other results of washed bloods at a later time. The discrepancies for the results in Figure 5 are within experimental uncertainties after about 11 minutes, ensuring that the roles of clotting factors and platelets are minor in the hemostatic enhancement associated with chitosan. The effect of pH decreases as time increases. 

## 4. Discussion

### 4.1. Aspects of the Hemostatic Characteristics of Chitosan Gauzes

In contrast to the time independent result of the pressure drop in Figure 2 for the flow of glycerol–water solution through gauzes, Figure 3, Figure 4 and Figure 5 show the time-dependent nature of the increase in pressure drop together with the differences of pressure drops across different gauzes for blood flows. These indicate that the phenomenon of hemostasis involves an interaction between blood (mainly RBCs, the red blood cells) and gauze materials, and the effects of such an interaction should be accumulative. Two plausible consequences of the interaction are: (i) the increase in blood viscosity as a result of the increase in hematocrit due to water absorption, which occurs for all absorbent gauzes, and (ii) the blockage effect due to the accumulation (adhesion and aggregation) of RBCs on the gauzes, which is particularly related to chitosan-based gauzes [6,21]. The order for better water absorption, “cotton gauze > non-woven gauze > rayon gauze”, could be suggested from the result of Figure 3 as the blockage effect due to RBCs accumulation is not significant for those three gauzes [21]. However, it is not clear whether the cotton or the chitosan gauze—A1 (made of chitosan and rayon fibers) has a better water absorption characteristic, as the water absorption ability of cotton is less than that of chitosan [34] but is better than that of rayon. The chitosan gauze—A1 has a better hemostatic performance than cotton and the other two gauzes because it possesses an additional blockage effect due to the accumulation of RBCs.

Based on the findings from the literature (as stated in the third paragraph of Section 1) and the results in Figure 3, Figure 4 and Figure 5 above, a rough picture for the process of hemostasis using chitosan gauzes is summarized as below. When a dressing consists of chitosan-based gauzes is applied at a bleeding wound subject to an “opposite applied pressure” (exerted manually, through packing, or using tourniquets), the blood flow through the gauzes is slowed down and becomes denser (the hematocrit, and thus, the blood viscosity, increases [33]) rapidly in time because of the high water absorption of the gauzes. Meanwhile, the RBCs next to the chitosan yarns adhere to them due to the attractive Coulomb force because they are of opposite charges, coating one or, at most, several layers of RBCs on the gauze. At the same time, H^+^ ions are released continuously from chitosan (associated with the NH_3_^+^ molecules [2]) into the blood, reducing the negative charges on the surrounding RBCs (associated with the COO^−^ molecules [22]), and thus, the repulsive electric double layer force between RBCs. Consequently, the RBCs in the neighborhood of the gauze aggregate among themselves, and adhere to the layer(s) previously coated on the gauzes. The aggregation process continues as more H^+^ ions are released, forming a layer-by-layer structure [21], narrowing the grid space in the gauze in time, and finally, the “mucoadhesive barrier” is formed and the bleeding is blocked. Such a blocking mechanism can be facilitated more easily in practice as dressings are made of layers of gauzes, and those gauze layers are placed together in a more or less “staggered” pattern in general. It takes about several minutes for the success of the above hemostatic processes in practice [6]. In summary, the above processes associated with hemostatic enhancement due to chitosan are primarily of a mechanical nature, involving some chemistry, but are essentially independent of classical clotting pathways, as suggested qualitatively through animal tests and medical treatments [9,10,11,12,13], and demonstrated quantitatively by the results in Figure 5. 

The above explanation for the formation of the layer-by-layer structure and further, the “mucoadhesive barrier”, is consistent with the results in Figure 4. The hemostatic performance (proportional to pressure drop) of the chitosan gauze—A2 is better than that of the chitosan gauze—A1 because the former contains more chitosan materials (a larger base weight), so that more H^+^ ions can be released. Similarly, the acidified gauze has more H^+^ ions on it, and thus, performs better than the non-acidified gauze in hemostasis. By comparing the difference for the results between A1 and A2 gauzes and that between the A1 and NA1 gauze, the acidification is a more effective factor for improving hemostasis in chitosan-based gauzes, in comparison with that using more chitosan materials (corresponding to a larger base weight). In the application of chitosan-based gauzes in practice, it would be better using acidified gauze layers. Gauzes with a higher base weight are required if the space for hemostasis is limited; they are not necessary in general as a same hemostatic performance can be achieved by using more layers of gauzes [6] with less base weight. 

More effective wound gauzes could be designed based on the above results and discussions. For example, the chitosan-based gauzes in this study would perform better if the rayon fibers for forming composite chitosan-based yarns were replaced by other fibers with better water absorption properties and if the chitosan-based gauzes were acidified to a lower pH. The former proposal is to increase the viscosity by enhancing the water absorption property of the gauze, while the latter is to increase the RBC aggregation by releasing more H^+^ ions from the gauzes into the flowing blood.

The pressure drop associated with the result for whole blood–2 (with certain amounts of NaOH added) is less than that for whole blood–1 in Figure 5 initially (say, for the first ten minutes), which could be explained that part of the H^+^ ions released from chitosan interact with OH^−^ from NaOH, instead of interacting with COO^−^ on RBCs. As the amounts of OH^−^ from NaOH are limited in comparison with the amounts of COO^−^ on RBCs, the result of whole blood–2 finally approaches that of whole blood–1. 

### 4.2. Flow-Through Devices

In comparison with many static tests in the literature, the study of the hemostatic performance of gauzes using flow-through devices (such as the Constant Flow Rate (CFR) device here and the Constant Pressure Gradient (CPG) device in Appendix A and in the literature [28]) is more realistic, as the performance was assessed in a flowing blood environment. Furthermore, the fabrication and the operation of the flow-through devices are simple, and thus, the devices are also convenient tools for designing better gauzes and for studying some hemostatic properties of gauzes using different blood samples, such as “whether the hemostatic enhancement due to chitosan is independent of the classical clotting pathways” in this study. 

The hemostatic performance for a given gauze assessed in the flow-through device is relative, and the absolute values of the pressure drops measured are for reference. For example, one can tell that the chitosan gauze—A1 is better than the cotton gauze according to the results in Figure 3, because the pressure drop across the chitosan gauze—A1 is higher. However, the value of the pressure drop could be increased if more layers of gauzes were inserted in the gauze chamber in Figure 1a. Thus, Jesty et al. [28] defined a hemostatic index in their CPG device for discriminating the relative performance of various gauzes. Following the same idea as Jesty et al., we can also define a hemostatic index (HI) here for a given gauze relative to the performance of the standard cotton gauze as
(1)$$\mathrm{HI}=\frac{\mathrm{Pressure}\;\mathrm{drop}\;\mathrm{across}\;\mathrm{given}\;\mathrm{gauze}}{\mathrm{Pressure}\;\mathrm{drop}\;\mathrm{across}\;\mathrm{the}\;\mathrm{cotton}\;\mathrm{gauze}}.$$

The hemostatic indexes for different gauzes in the present study can be calculated using the results in Figure 3, Figure 4 and Figure 5, and the HIs for several gauzes are shown in Figure 6 for illustration. A gauze is better than the standard cotton gauze if HI > 1 or vice versa. Both chitosan-based gauzes are better than the cotton gauze, while the rayon and non-woven gauzes are worse. The values of HI still vary with time in Figure 6, but their variations are smaller in comparison with the values of pressure drops in Figure 3, Figure 4 and Figure 5. One can discriminate the hemostatic performances of various gauzes more easily and definitely using the results of HIs, instead of using the results of pressure drop.

According to the qualitative agreement between the results in Figure 3 and Figure 4 for the CFR device and those in Figure A2 (see Appendix A) for the CPG device, both devices are capable for assessing dynamically the hemostatic performance of gauzes/dressings. In general, the CFR device is smaller and the amount of test fluid (blood) required is substantially less than that of the CPG device. Consider the amount of blood required for passing through the gauzes in the tests. For example, 16.8 mL of blood was required for the tests of those four gauzes in Figure 3 using the CFR device, if the tests (one for each gauze) were performed for 10 minutes. On the other hand, 21.1 and 41.5 mL were required for the corresponding four tests in Figure A2a (for a driving pressure head 60 mmHg) and Figure A2b (for 120 mmHg), respectively, using the CPG device. These results imply that the test fluid required for the CFR device is about 20% (or 60%) less than that required for the CPG device for a driving pressure head 60 mmHg (or 120 mmHg). Furthermore, as the size of the CPG device is greater than that of the CFR device, more blood is required for filling the device for the experiment in the CPG device. However, the filling test fluids could be reduced substantially for the CPG device if the size of the intermediate chamber (or the plasma chamber in reference [28]) in Figure A1 is large enough to store the total amount of blood sample for the requested test. A buffer liquid with density less than that of the blood can be filled in the flexible tube and the upper reservoir above the intermediate chamber of Figure 1a for exerting the required applied pressure gradient for the blood flow. In such a case, the upper reservoir should be raised further to a higher place to take into account the density difference between the buffer and the test liquid.

However, the CPG device has an advantage over the CFR device that the process of bleeding and hemostasis can be simulated better using the CPG device, as realistic driving pressure heads (say, 120 mmHg) can be applied, and the blood flow can finally be stopped by appropriate gauzes (refer to Figure A2d of Appendix A). On the other hand, the flow rate in the CFR device is set constant throughout the test; the blood flow cannot be brought to a stop through the gauze. Thus, the CFR device is an alternative, not a replacement, of the CPG device, for assessing dynamically the hemostatic performance of gauzes. Both the CPG and CFR devices have their own advantages and disadvantages for applications. 

### 4.3. Remarks on the Effect of Molecular Weight and Deacetylation Degree of Chitosan

The molecular weight and deacetylation degree of chitosan are important factors for hemostatic application using chitosan, as discussed in the literature. However, they are not the primary parameters concerned in the present study. Two main findings of this study are: (1) hemostatic enhancement via chitosan is independent of classical clotting pathways, and (2) the hemostatic performance of an acidified gauze is better than that of a non-acidified gauze. Those two findings remain unchanged for a chitosan-based gauze with given values of molecular weight and deacetylation degree (and thus, fixed numbers of an amino group). Here, the acidification process is to immerse the finished chitosan-based gauze into acid alcohol for about 30 minutes, changing the amino group to the ammonium group, or imparting more H^+^ ions on the gauze (under fixed molecular weight and deacetylation degree). As a result, more ions can be released when the gauze is exposed to the blood, enhancing RBCs aggregation, and thus, hemostasis. Hemostatic performance would be different for gauzes with different values of molecular weight and deacetylation degree, which deserves further study for a more effective chitosan-based wound gauze. 

## 5. Conclusions

A flow-through device under constant flow rate was implemented and demonstrated successfully for studying dynamically the hemostatic characteristics of various wound gauzes by measuring the pressure drops across the gauzes, with less test blood (20%–60% less than that using a device driven by a constant pressure gradient associated with a pressure head at 60–120 mmHg). Experiments were performed using both the whole blood and washed blood (with clotting factors and platelets removed), and the results agree with each other within 10% discrepancy. Such a result indicates that the hemostatic enhancement due to chitosan is essentially independent of classical clotting pathways and suggests that there is direct interaction between RBCs and chitosan for hemostasis. An explanation was provided primarily from the mechanical viewpoint for the aspect of enhancing hemostatic performance via chitosan, which could be helpful for understanding the phenomena and designing more effective wound gauzes. The hemostatic performance of chitosan-based gauzes can be enhanced effectively through the acidification of the gauze. The acidified/non-acidified chitosan gauze is about 60%/30% better than the standard cotton gauze, according to the result of hemostatic indexes in Figure 6.

## Figures and Tables

**Figure 1 polymers-12-02391-f001:**
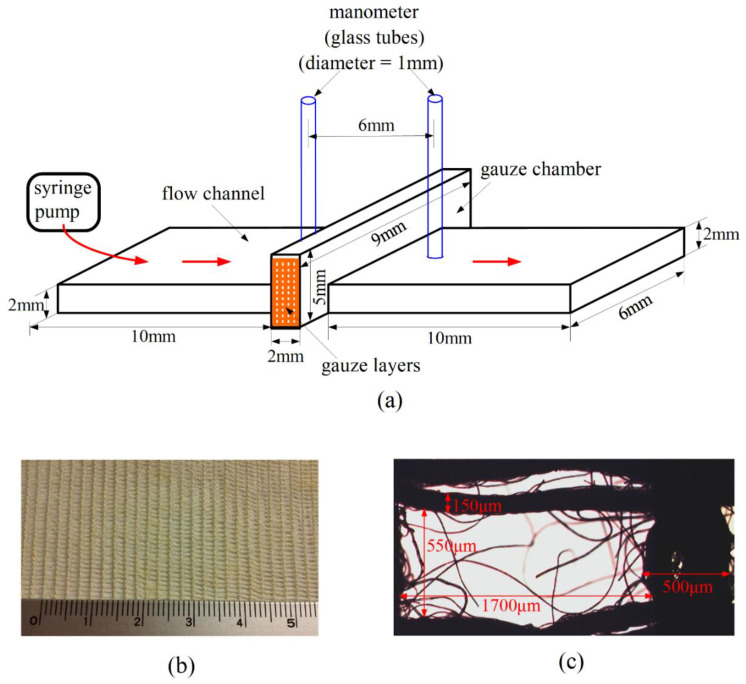
(**a**) Sketch of the Constant Flow Rate (CFR) device. The scales in the figure are referred to the inner scales of the chamber, the tubes, and the channel. (**b**) A chitosan gauze—A1. The numbers in the ruler refer to the scales in centimeter. (**c**) A typical mesh of the gauze, with length scales indicated.

**Figure 2 polymers-12-02391-f002:**
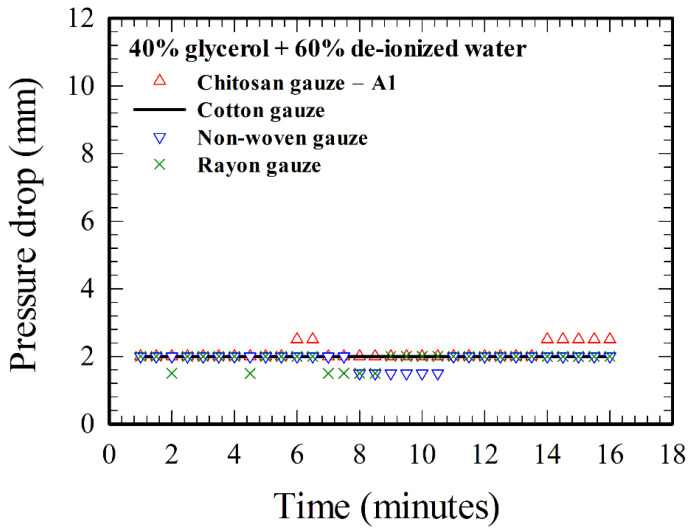
Time evolution of the pressure drops across various gauzes for glycerol–water solution.

**Figure 3 polymers-12-02391-f003:**
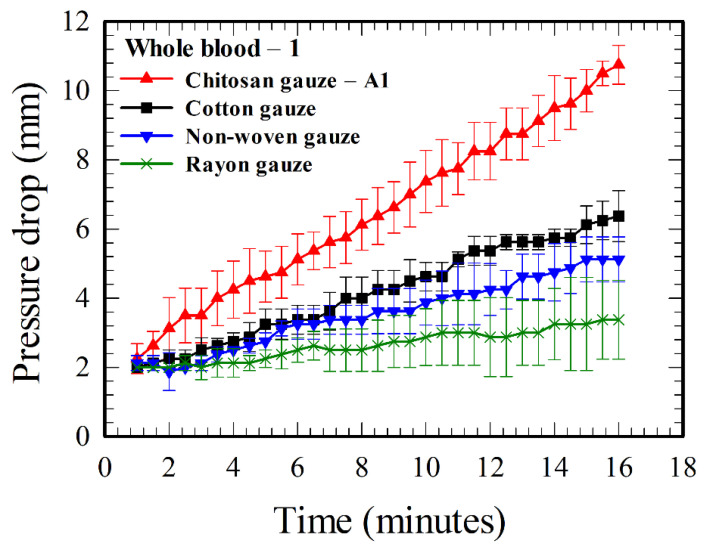
Time evolution of the pressure drops across various gauzes for whole blood–1.

**Figure 4 polymers-12-02391-f004:**
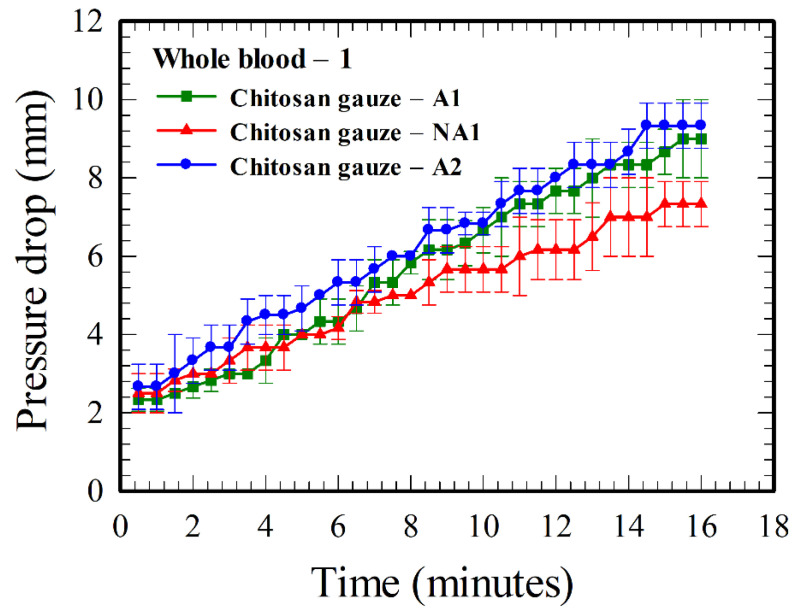
Time evolution of the pressure drops across various chitosan-based gauzes (refer to Table 1) for whole blood–1. Both A1 and A2 gauzes were acidified, while NA1 gauze was not. The base-weight of the A2 gauze is larger than that of the A1 gauze.

**Figure 5 polymers-12-02391-f005:**
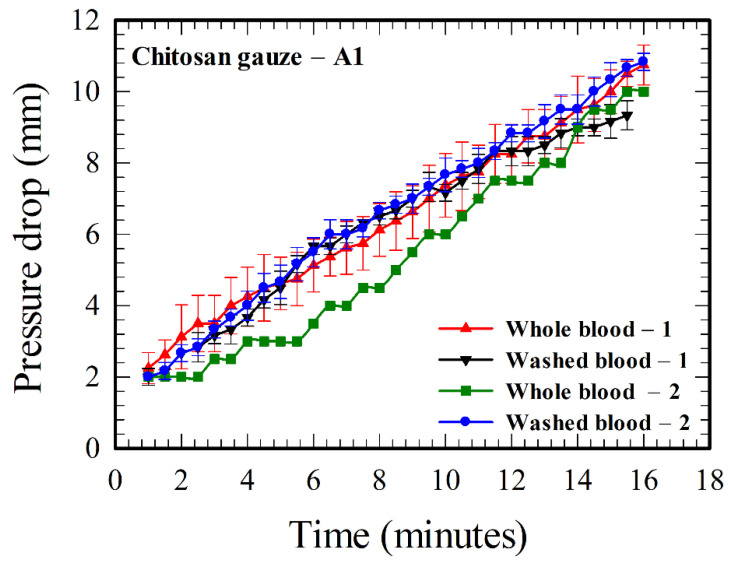
Comparison of the pressure drops across the chitosan gauze—A1 between the flows of whole and washed bloods at different values of hematocrit and pH (refer to Table 2).

**Figure 6 polymers-12-02391-f006:**
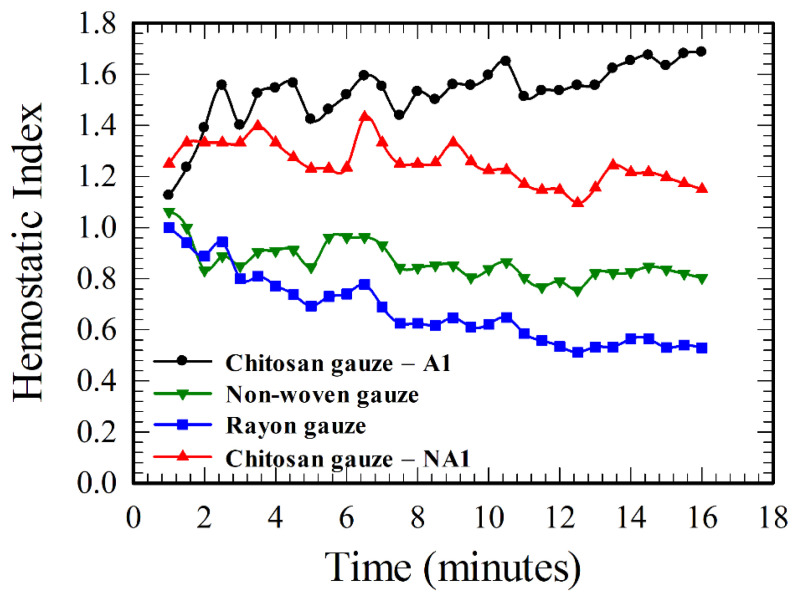
Comparison of the hemostatic indexes for different gauzes.

**Table 1 polymers-12-02391-t001:** Gauzes in this study.

Gauze	Average Mesh (μm × μm)	Acidification ^#^
Chitosan gauze—A1	1700 × 550 (85 g/cm^2^) *	Yes
Chitosan gauze—A2	350 × 350 (166 g/cm^2^) *	Yes
Chitosan gauze—NA1	750 × 600 (130 g/cm^2^) *	No
Cotton gauze	1200 × 550	No
Non-woven gauze	1250 × 250	No
Rayon gauze	2150 × 950	No

* The value inside the parenthesis is the base weight of the gauze. **^#^** Acidification refers to immersing the gauze in an acid alcohol [30].

**Table 2 polymers-12-02391-t002:** Blood samples in this study.

Blood Sample	Contents	pH
Whole blood–1	Human blood + anticoagulant	6.60
Whole blood–2	Human blood + anticoagulant + NaOH	7.52
Washed blood–1	Washed RBCs (40%) + Saline (60%)	7.08
Washed blood–2	Washed RBCs (50%) + Saline (50%)	7.55

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
