# Peer review of "Hemostatic Enhancement via Chitosan Is Independent of Classical Clotting Pathways—A Quantitative Study"

_polymers, 2020, doi:10.3390/polym12102391_

Round 1
Reviewer 1 Report
Peer review report on "Hemostatic Enhancement via Chitosan is Independent of Classical Clotting Pathways – A Quantitative Study."
- Recommendation
Major corrections
- Comments to the author:
Manuscript ID: polymers-941867
Type of manuscript: Article
Title: Hemostatic Enhancement via Chitosan is Independent of Classical Clotting Pathways – A Quantitative Study.
Overview and general recommendation:
In the present work, it was proposed the uses of Chitosan gauze in a continuous flow-through device with the test blood flowing through the gauze sample at a constant flow rate and the pressure drop across the gauze measured, for assessing the hemostatic performance of the gauze. The study is significant because it proposes a new device (smaller and more convenient than those already available), and also because measured in a dynamic way the homeostasis capacity of the chitosan gauze in comparison with others.
I found the paper with a high novelty degree and generally well-written and supported. However, some issues need to be addressed before publication can proceed.
In the abstract section, it is necessary to include quantitative data to support the conclusions. For example, lines 20-22 claim that "…and their results agree with each other, indicating quantitatively that the hemostatic enhancement of chitosan is indeed independent of classical clotting pathways, which was demonstrated qualitative through animal tests in the literature…". How the results agree with each other? How is it quantitively demonstrated the hemostatic enhancement? And that is independent of classical clotting pathways? Please explain better using quantitative data and conclusions.
Lines 23-27: It is necessary to elaborate better: "The proposed device and method can be applied for evaluating quantitatively the hemostatic performance of various gauzes in a flowing blood environment (in comparing with many existing static tests) with less test blood (in comparing with the existing device driven by a constant pressure head), and thus help understand the associated hemostatic phenomena and design better-wound gauzes." It is better to demonstrate qualitatively the quantitative evaluation and how that improves the actual methods and results available with existing devices. Finally, it elaborates better how that helps understand the associated hemostatic phenomena.
It is better to order keywords in alphabetical order.
It is necessary to review the introduction section for English grammar and punctuation signs. In general, there is enough background available to understand the concepts and previous work, but it is possible, reducing some text will help for a better understanding. On the other hand, the literature is old. It is required to add more recent references.
Line 134: Please check ml/hr should be mL/h.
Line 138: Remove explanations of the performance. That should be explained in the discussion section.
Lines 150-152: It is not enough to report: "…the chitosan powder with a molecular weight of 300-500 kDa and deacetylation level more than 75% was first dissolved into 0.5 M acetic acid solution (purity: 100%, J. T. Baker, United 152 States)." Those are technical data provided by the supplier. However, the properties of chitosan are very dependent on molecular weight and deacetylation degree. Therefore, the absence of these parameters in the characterization is unacceptable. The European Chitin Society does not recommend publication of the research papers on chitosan, in which this information is missing. I recommend providing data of a proper characterization (for example, Mv using the Mark-Houwink Sakurada equation and using the appropriate literature to calculate the Mv).
When indicate %, please indicate what kind of % is it.
I suggest adding a scheme to explain the fabrication of the chitosan gauzes. That will facilitate the readers to understand better the manufacturing process.
All the equipment and reagents must be referenced as (manufacturer, city, country) without exception.
Is the methodology for washed blood a developed procedure or a literature methodology? It is from literature. Please cite the reference correctly.
Lines 202-203: How do you know that is the viscosity of the human blood? Cite the proper reference.
Lines 205-208: Explain the drops of 0.5mm for Chitosan gauzes.
Explain why cotton and non-woven gauzes have those drop pressures. Why do they have that ability?
Lines 243-246: Why the reader must understand that the results agree with each other with the actual evidence? Please elaborate.
Lines 299-302: Explain the statement: "…The above processes associated with hemostatic enhancement due to chitosan are primarily mechanical, involving some chemistry..."
Lines 304-307: Explain better the statement: "…The hemostatic performance (proportional to pressure drop) of the chitosan gauze - A2 is better than that of the chitosan gauze – A1 because the former contains more chitosan materials (a larger base weight) so that more H+ ions can be released." What does it mean that more H+ ions can be released in the present conditions?
Lines 307-308 also need more explanation. A more in-depth analysis of the acidification process in front of the hemostatic process needs to be elaborated.
Lines 308-314: I consider crucial to explain based on a confident analysis of the deacetylation degree, the importance of the acidification process, and for the hemostatic effect. It is not only the molecular weight, the reason for better performance. It should be derived mainly from the deacetylation degree as well.
Conclusions should be elaborated in terms of the quantitative results. Introduce the drop pressure obtained and the HI index to conclude the better performance.
Author Response
See the attached file, please!

Reviewer 2 Report
Topic of the study as well as obtained results are very interesting and manuscript is quite well written. However, there are present some minor points which decrease the overall quality of the manuscript:
1. chapter 2.2, acetic acid solution: according to the guide for authors, there should be mentioned city of the origin and for suppliers from US also state.
2. page 10, fig. 5: The points shouldn't be connected here.
3. page 10, fig. 5: it should be fig 6, as the fig 5 is already on page 8. In the text (for example lines 345 or 348) it is correct.
Author Response
See the attached file, please!

Round 2
Reviewer 1 Report
Several suggestions were addressed accordingly. However, the molecular weight and deacetylation degree (at the end represents the number of amine groups that can accept protons under acidic pH) is important if it has an important role in the application. However, according to the comments, the MW and the DD are not important in the present study. It seems that any chitosan (despite this fact is not proven in the present study) with any DD and MW can have the same response in the GAUZE. Changes in pH with dissolved chitosan imply cationization of Amine groups (-NH2 to -NH3+), which relies on the number of available amine groups (which also depends on the deacetylation degree).
Author Response
See the attached file, please!
